# Quantifying Understory Vegetation Cover of *Pinus massoniana* Forest in Hilly Region of South China by Combined Near-Ground Active and Passive Remote Sensing

**Ruifan Wang** [1,2], **Tiantian Bao** [1,2], **Shangfeng Tian** [1,2], **Linghan Song** [1,2], **Shuangwen Zhong** [1,2], **Jian Liu** [1,2], **Kunyong Yu** [1,2] and **Fan Wang** [1,2,*]

1   College of Forestry, Fujian Agriculture and Forestry University, Fuzhou 350000, China
2   University Key Laboratory for Geomatics Technology and Optimize Resource Utilization in Fujian Province, Fuzhou 350000, China
*   Correspondence: 000q817013@fafu.edu.cn; Tel.: +86-182-5195-2597

**Abstract:** Understory vegetation cover is an important indicator of forest health, and it can also be used as a proxy in the exploration of soil erosion dynamics. Therefore, quantifying the understory vegetation cover in hilly areas in southern China is crucial for facilitating the development of strategies to address local soil erosion. Nevertheless, a multi-source data synergy has not been fully revealed in the remote sensing data quantifying understory vegetation in this region; this issue can be attributed to an insufficient match between the point cloud 3D data obtained from active and passive remote sensing systems and the UAV orthophotos, culminating in an abundance of understory vegetation information not being represented in two dimensions. In this study, we proposed a method that combines the UAV orthophoto and airborne LiDAR data to detect the understory vegetation. Firstly, to enhance the characterization of understory vegetation, the point CNN model was used to decompose the three-dimensional structure of the *pinus massoniana* forest. Secondly, the point cloud was projected onto the UAV image using the point cloud back-projection algorithm. Finally, understory vegetation cover was estimated using a synthetic dataset. Canopy closure was divided into two categories: low and high canopy cover. Slopes were divided into three categories: gentle slopes, inclined slopes, and steep slopes. To clearly elucidate the influence of canopy closure and slope on the remote sensing estimation of understory vegetation coverage, the accuracy for each category was compared. The results show that the overall accuracy of the point CNN model to separate the three-dimensional structure of the *pinus massoniana* forest was 74%, which met the accuracy requirement of enhancing the understory vegetation. This method was able to obtain the understory vegetation cover more accurately at a low canopy closure level ($R_{low}^2 = 0.778$, $RMSE_{low} = 0.068$) than at a high canopy closure level ($R_{High}^2 = 0.682$, $RMSE_{High} = 0.172$). The method could also obtain high accuracy in version results with $R^2$ values of 0.875, 0.807, and 0.704, as well as RMSE of 0.065, 0.106, and 0.149 for gentle slopes, inclined slopes, and steep slopes, respectively. The methods proposed in this study could provide technical support for UAV remote sensing surveys of understory vegetation in the southern hilly areas of China.

**Keywords:** southern hilly region; *Pinus massoniana* forest; understory vegetation cover; airborne LiDAR; UAV; deep learning

## 1. Introduction

Understory vegetation is commonly defined as the collection of vegetation with distinct stratification and spatial heterogeneity under the tree canopy in a forest system, including shrubs, herbaceous vegetation, and mossy vegetation [1,2]. In hilly areas in southern China that are experiencing soil erosion (i.e., where the vertical structure is relatively simple), the understory vegetation largely consists of herbaceous plants [3]. Understory vegetation plays a significant role in maintaining ecosystem stability (e.g.,

carbon sinking capacity as well as soil and water conservation) [4–9]. Understory vegetation cover can generally be framed as the percentage of vertical projection of roots, stems, and leaves of all understory vegetation layers in a unit area study unit [10]. As a criterion used as a measure of ground cover, it can be used as an indicator of the soil erosion status of an ecosystem [11].

With the development of quantitative remote sensing technology, near-ground remote sensing systems with their ability to identify features with high accuracy are gradually becoming the main means of monitoring the state of forest ecosystems at a fine scale [12–14]. An active remote sensing system is a system in which the sensor emits electromagnetic radiation (at a certain frequency) towards a target, and the radiation then bounces off the target and is received by the system for processing, providing physical information about the target. The Light Detection and Ranging (LiDAR) system, as an emerging active remote sensing system, is widely used in 3D information acquisition and 3D structure modeling of land features, and has become an important means of quantifying forest structure due to characteristics such as high data accuracy, strong canopy penetration ability, and no interference from sunlight and shadows [15–18]. Simultaneously, the laser beams released by UAV LiDAR sensors can penetrate the forest canopy gaps in the forest system, which is particularly useful in depicting the understory vegetation structure [19–21]. However, cluttered point cloud information, the traditional height stratification method, makes it difficult to directly characterize the texture and attributes of understory vegetation in undulating terrain [22], limiting the widespread use of LiDAR in excavating forest information in areas such as the southern hilly region of China.

Improvements in computing technologies have enabled numerous innovations in deep learning methods, especially when computer vision is concerned [23–25]. Semantic segmentation is an image processing technology based on a deep learning platform, and it makes an outstanding contribution in the mining of green volume spatial distribution information and quantifying green parameters [26–28]. However, LiDAR point cloud data are a high-dimensional representation of 2D image data, making it not feasible for direct use when applying image semantic segmentation algorithms [29]. As a three-dimensional generalization of the convolutional neural network, the point CNN model inherits the network and CNN method to obtain features [30,31]. However, to accommodate the spatially disordered nature of point cloud data, the X-Conv operator is introduced to weigh and permute the feature matrix after the point cloud convolution transformation. During execution, the model can also ensure high stability of the feature shape and structure of the point cloud [30], which makes it an ideal model for UAV LiDAR vegetation point cloud data decomposition and undergrowth information enhancement [32].

Within the context of vegetation cover, understory vegetation is considered a two-dimensional indicator [10]. Therefore, it is difficult to reflect "vertical projection" in the conceptualization of understory vegetation cover using enhanced understory information from the UAV laser point cloud. Therefore, two-dimensional images are needed as a basis for calculating the understory vegetation cover. On the one hand, as a representative of near-ground passive remote sensing systems, the UAV remote sensing system can provide fine vegetation texture and spectral information of the sample site within the orthophoto [33–35]. On the other hand, UAV oblique photography can obtain information of one feature from different angles, and do so simultaneously, effectively avoiding obstacle occlusion and greatly increasing the accuracy of the basic data for quantifying understory vegetation parameters [36–38]. However, UAV oblique photography also causes a systematic deformation of vegetation morphology in the photos, and there is no effective method to correct the vegetation morphology in the two-dimensional images; therefore, the UAV oblique photography images cannot guarantee the effective representation of "vegetation vertical projection area" and cannot be used to accurately quantify the forest vegetation cover. Therefore, orthophotos should be used as a reference to extract the vertical projected area of vegetation [39].

The structure and properties of the forest can be obtained precisely by combining the UAV LiDAR remote sensing system and the UAV orthophoto, as the former can provide the spatial pattern of the ground, while the latter contains the texture of the ground [40–42]. Nevertheless, data collected from the two different systems were not sufficiently matched within the sample plots, resulting in an abundance of understory vegetation information in the sample region that was not reflected in the two dimensions. The multi-source data synergistic mechanism has not been fully revealed in the remote sensing quantification of understory vegetation cover, which leads to active and passive remote sensing not exerting their full potential when quantifying the corresponding understory vegetation parameters. The point cloud back projection algorithm can implement the two-dimensional transformation of point cloud information within the sample plot and then assign the enhanced spatial information of the point cloud onto the two-dimensional image. With the help of this method, the UAV LiDAR remote sensing system and the UAV orthophoto can be combined effectively, and the characterization of stand structure and understory vegetation can be comprehensively depicted [10].

In this study, Hetian Town in Changting County was used as the study area, and UAV orthophotos and UAV LiDAR were used as representatives of near-ground active and passive remote sensing systems. Using the point CNN model framework and point cloud back projection algorithm, our objectives were (1) to obtain a high-precision three-dimensional structure of forest stands in the southern hilly areas of China; (2) to develop a remote sensing estimation method for understory vegetation cover by combining active and passive remote sensing systems; and (3) to explore the mechanisms underlying the remote sensing estimation of understory vegetation cover. The results of this study are of great practical significance for the rapid monitoring of local soil erosion status and can also provide a reference for the precise management of soil erosion in forests.

## 2. Study Area and Sample Site Overview

Hetian Town, Changting County (25°35′–25°46′ N, 116°16′–116°30′ E), with a total area of 296 km$^2$, is located southwest of Fujian Province. The area is surrounded by mountains, has undulating terrain, and is mostly covered with red soil. The average altitude of the study area was 390 m, and the average slope of the hills in the study area was calculated to be approximately 20° via geostatistical analysis. Enclosed by a network of rivers, the climate is mild and rainfall is abundant [43]. The vertical forest structure in the study area is relatively homogeneous, with *Pinus massoniana* as the main tree species, and the understory is dominated by the herbaceous vegetation of *Dicranopteris dichotoma* [36,44].

In this study, 60 sample plots of size 20 × 20 m were constructed. The four corner points of the sample plots were located using the DJI Phantom 4 RTK system with a positioning accuracy of centimeters. The distribution of the experimental plots is shown in Figure 1.

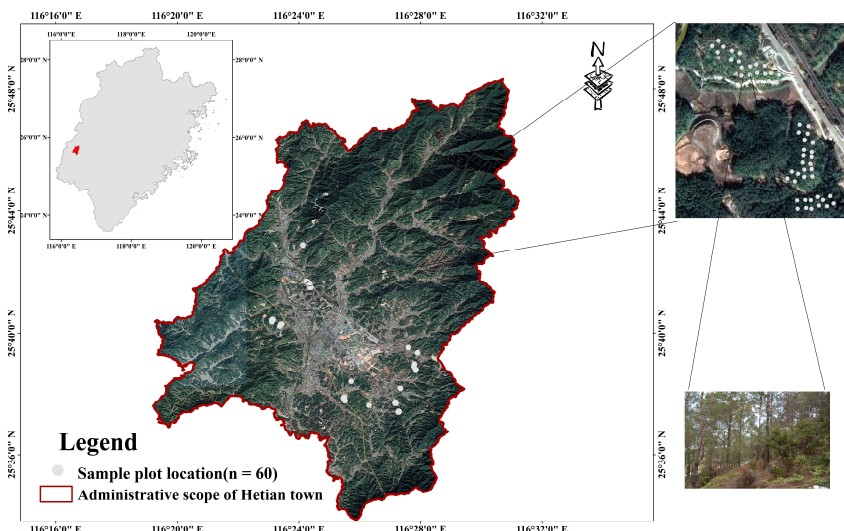

**Figure 1.** Schematic diagram of experimental plot.

## 3. Data and Methodology

### 3.1. Data Acquisition

#### 3.1.1. Field Measurements Acquisition

Photographs of the understory vegetation quadrangles (east, south, west, and north) were taken every 5 m along the diagonal of the sample plots at approximately 1.5 m from the ground using the Samsung SM-G9910 sensor [45]. Then, these photographs were used to calculate the actual ground value of understory vegetation cover. Each image was 6000 pixels in length and 4000 pixels in width. All data were collected between May and June 2022.

#### 3.1.2. UAV Visible Light Remote Sensing Data Acquisition

A DJI M300 UAV equipped with a P1 visible-light camera was used as the flight platform to capture orthophotos of the sample sites in orthogonal flight mode during windless and light-filled hours (AM 9:00–PM 2:00). These UAV visible light photos were used to stitch UAV orthophoto images. The flight height varied with the canopy closure of the sample site in the range of 100–150 m. The overlap rate between the flight heading and side directions was set to 80%. To ensure there was no evident deformation of the image features within the boundary of the sample site, the boundary was extended by 15 m to represent the actual flight range. The photographs were spatially processed and modeled using DJI Terra software (https://www.dji.com/cn/dji-terra) to generate a digital orthophoto map (DOM) in June 22. The spatial resolution of the orthophoto was 0.05 m, and the coordinates were projected onto WGS_1984_UTM_Zone_50N.

#### 3.1.3. UAV LiDAR Data Acquisition

Synchronized with the acquisition time of the UAV visible light remote sensing data, the M300 UAS equipped with the L1 laser lens was used to acquire LiDAR 3-echo point cloud data using the waypoint hovering mode. UAV airborne LiDAR data were used to probe the three-dimensional structure of the forest. The laser spot size of this lens was 52 mm × 491 mm, the 3-echo pulse scanning frequency was 160 KHz, the scanning angle was set to a maximum of $\pm 30°$, and the rest of the UAV flight parameters were the same as those used in the acquisition of visible light data. The M300's built-in RTK provides a maximum positioning accuracy of 1 cm + 1 ppm on plane and 1.5 cm + 1 ppm on elevation, and high-precision data files were obtained with the help of the GNSS, INS, IMU, and inertial guidance system. The acquired point-cloud density exceeded 600 points/m$^2$. Finally, the data were stored in the las 1.4 standard format to store x, y, z values, echo numbers, custom classifications, etc. The point cloud data were projected onto the same frame as the visible light data. The final data volume was approximately 10 GB.

### 3.2. Methodology

First, the UAV visible photos and airborne LiDAR data of the *Pinus massoniana* forest sample sites were acquired, and all remote sensing data were confirmed to be within the WGS_1984_UTM_Zone_50N geographic coordinate frame. Second, the UAV visible light photos were used to stitch the UAV orthophoto images, and the point cloud information was obtained using airborne LiDAR data. Third, the point cloud data were pre-processed, and a semantic segmentation model was constructed for the *Pinus massoniana* forest. This was performed based on which the three-dimensional structure of the forest stand was decomposed with high precision and the three-dimensional information of the understory vegetation was enhanced. Subsequently, the enhanced understory information was back-projected to the UAV orthophoto using the point cloud back-projection algorithm. Thereafter, the aggregated data were voxelized and binarized. The understory vegetation was quantified using binarized data. Meanwhile, based on the ground and canopy point sets obtained using semantic segmentation, the slope and canopy closure of the statistical sample sites were extracted and used as underlying factors. The near-ground photographs of the sample plots were taken, and the vegetation patterns were outlined using a threshold algorithm combined with a spatial interpolation algorithm to obtain the ground truth values of the understory vegetation. Finally, an accuracy analysis of the remote sensing estimation of understory vegetation under different slope and canopy closure conditions was performed. The experimental procedure is shown in Figure 2.

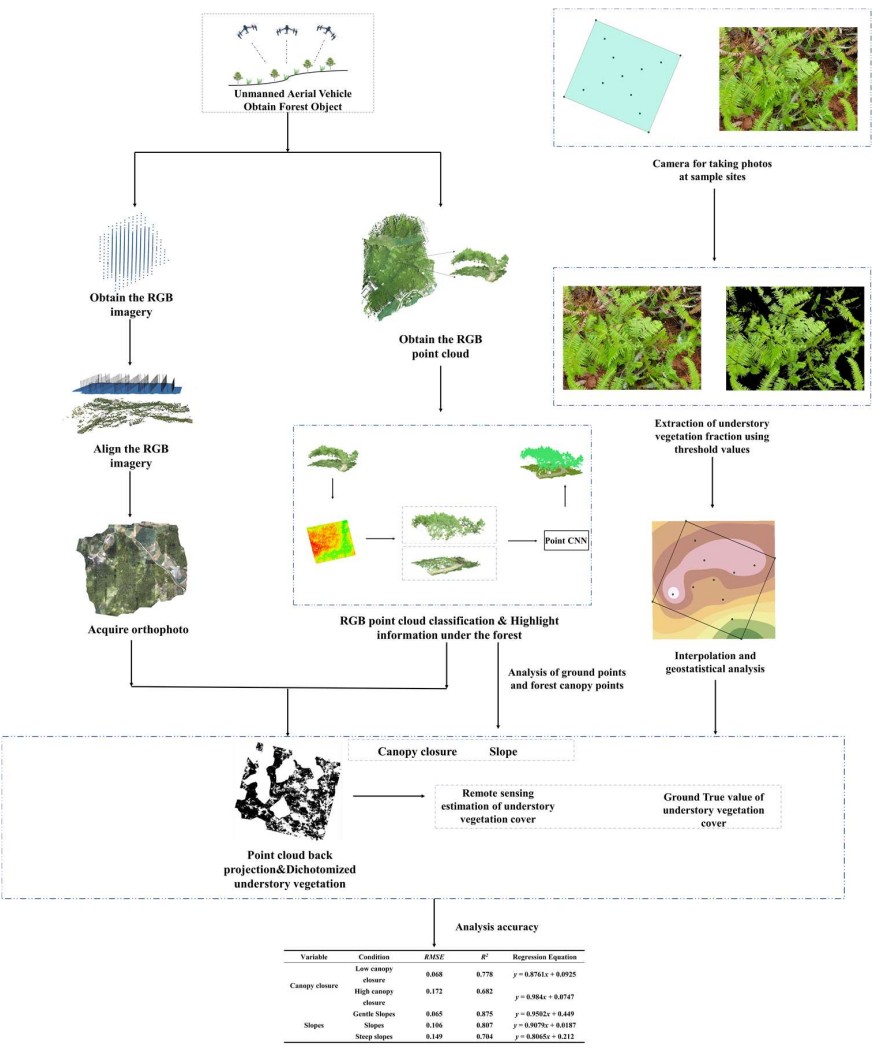

**Figure 2.** Flow chart of vegetation inversion method.

### 3.2.1. Data Pre-Processing

The LiDAR data were processed using LiDAR360 (https://www.lidar360.com/) software in June 25. The acquired raw LiDAR point cloud data were pre-processed through cropping and de-noising, and the built-in machine learning algorithm of the software was used to pre-classify the point cloud into three categories, namely: forest canopy, understory vegetation, and ground points. The machine learning classification tool of the software uses random forests to classify the point cloud data. By manually editing the categories of typical data in the same batch, the model was trained and subsequently batched to process a large amount of data. The classification results were further processed to construct a semantic segmentation dataset.

Hence, the preliminary classified point cloud data were loaded into the ArcGIS Pro2.9 deep learning environment and divided into $800 \times 800$ points tiles. Due to graphics card memory limitations, the batch size was set to four. The training categories were forest canopy, understory vegetation, and ground points. The point cloud was then finely tagged and classified using the point cloud tagging function, and the semantic segmentation dataset was built based on the fine classification results. Finally, the tag classification results were corrected using a visual interpretation check.

### 3.2.2. High-Precision Separation Method for Three-Dimensional Structure of *Pinus massoniana* Forest

The deep learning environment integrated into ArcGIS Pro2.9 includes the Point CNN architecture for high-accuracy classification of point cloud data. The model first selects points and the corresponding labels for the previous round of network input point sets, and then captures the information of the neighboring point sets of the sampled points using the K-neighborhood method, recursively convolveing the local network until all the point sets in the network are captured, and then a round of feature extraction is completed. After several rounds of training, the information loss rate converges at a certain point to obtain a semantic segmentation model, which can be used to complete the separation of the three-dimensional structure of the *Pinus massoniana* forest and the enhancement of understory vegetation information [32]. To prevent gradient explosion, the learning rate was set to 0.001 and the corresponding optimization method was chosen in this study.

Owing to the lack of spectral information in the acquired point cloud data, the echo intensity information was also trained in view of the data enhancement principle. The structure of the point CNN model is shown in Figure 3.

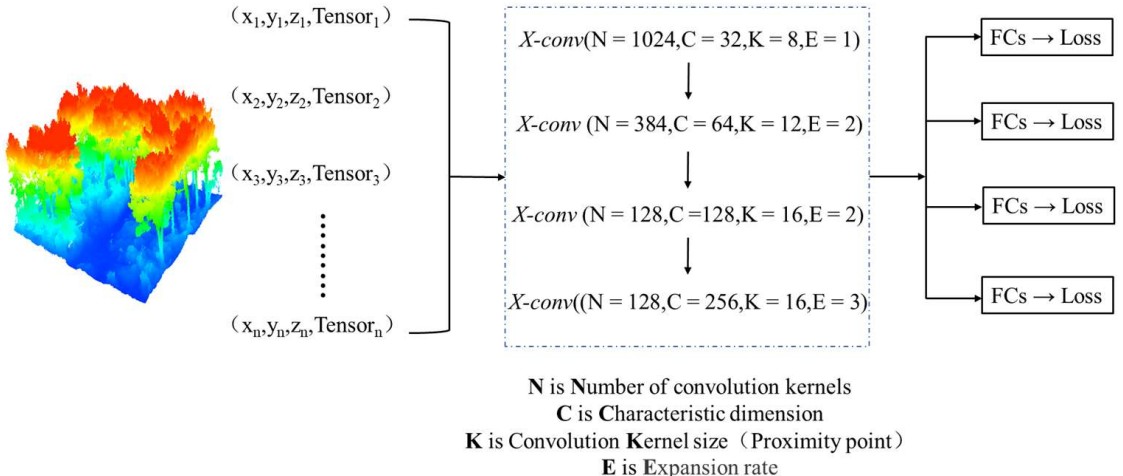

**Figure 3.** Point CNN model structure.

### 3.2.3. Two-Dimensional Presentation and Quantification of Three-Dimensional Information of Forest Understory Vegetation

As point cloud remote sensing images have linear constraints, point cloud 3D geographic coordinates can be back-projected to 2D image projection coordinates [9]. The API function embedded in PhotoScan (http://www.agisoft.cn/) was called via a Python script to automatically retrieve this transformation matrix and perform an inverse projection, thus projecting a collection of understory vegetation point clouds onto the UAV orthophoto in July 29. The mathematical expression for the inverse projection is

$$\begin{pmatrix} x \\ y \\ -h \end{pmatrix} = \mathbf{R} \cdot \begin{pmatrix} X \\ Y \\ Z \end{pmatrix} + \mathbf{T} = \begin{pmatrix} r_{11} & r_{12} & r_{13} \\ r_{21} & r_{22} & r_{23} \\ r_{31} & r_{32} & r_{33} \end{pmatrix} \cdot \begin{pmatrix} X \\ Y \\ Z \end{pmatrix} + \begin{pmatrix} t_1 \\ t_2 \\ t_3 \end{pmatrix} \tag{1}$$

where $x$ and $y$ are the horizontal and vertical coordinates of the raster data, respectively, $X$, $Y$, and $Z$ represent the 3D orientation of the point cloud data, $-h$ is the camera focal length, $\mathbf{R}$ is the angle conversion matrix, $\mathbf{T}$ is the translation conversion matrix, and $r_{ij}$ are the corresponding matrix parameters.

After projecting the undergrowth point cloud collection onto the UAV orthophoto, the point cloud was voxelized to the same resolution as the UAV image, and the understory and non-understory vegetation raster range were binarized. The understory vegetation cover was then calculated according to the definition of vegetation cover, that is, the ratio of the raster occupied by the understory vegetation to the total raster of the image. The formula used is as follows:

$$UVC = \frac{P_{UndVeg}}{P_{All}} \tag{2}$$

where $P_{All}$ is the photo image element and $P_{UndVeg}$ represents the image element occupied by the green vegetation in the photo.

### 3.2.4. Method of Calculating the Ground-Truthing Value of Understory Vegetation Cover

The HSV color space color extraction algorithm was called from the OpenCV library to outline the extent of understory vegetation in the field photos, so as to obtain the understory vegetation cover at a certain point. Then, the kriging interpolation tool provided by ArcGIS10.8 was used to interpolate the understory vegetation cover of all sampling points in the sample plots, and the geostatistical analysis module was used to accurately analyze the understory vegetation cover of the sample plots. The mean values of the sample sites were the analyzed using the statistical analysis module.

### 3.2.5. Method of Sample Site Information Statistics

The canopy and ground point clouds were extracted separately based on the semantic segmentation of the point cloud. The canopy point cloud was pixelated to the same resolution as the orthophoto of the sample site, the canopy extent and the sample site extent were binarized, and the ratio of the two could be identified as the sample site canopy closure. The ground point cloud of the sample site was converted into a point set, and the ground point set was interpolated with the sample site extent as the environmental boundary, and then the slope operation was performed to obtain the sample site factor. The above operations were all conducted in the ArcGIS10.8 environment.

## 4. Results

### 4.1. Field Measurements on Sample Site Topography, Canopy Closure, and Understory Vegetation Cover

Point cloud semantic segmentation was used to obtain canopy and ground point cloud collections. The corresponding methods were used to calculate the topography and stand cover of the sample sites, and the measured values of understory vegetation cover were calculated using the HSV color extraction algorithm based on the ground photo data of understory vegetation. The distribution of closure and topography of all the sample

plots in this study is shown in Figure 4 and Table 1. In the figure and table, UVCGround is the ground-truth value of the understory vegetation cover. It should be clarified that, unlike vegetation cover, the value of canopy closure was expressed as a multiple of 0.1 according to the relevant Chinese forestry standards. However, to make the data division more accurate, the value of canopy closure was kept at two decimals and rounded to the nearest half or whole number in this study. The slope of the sample site was maintained at three decimals.

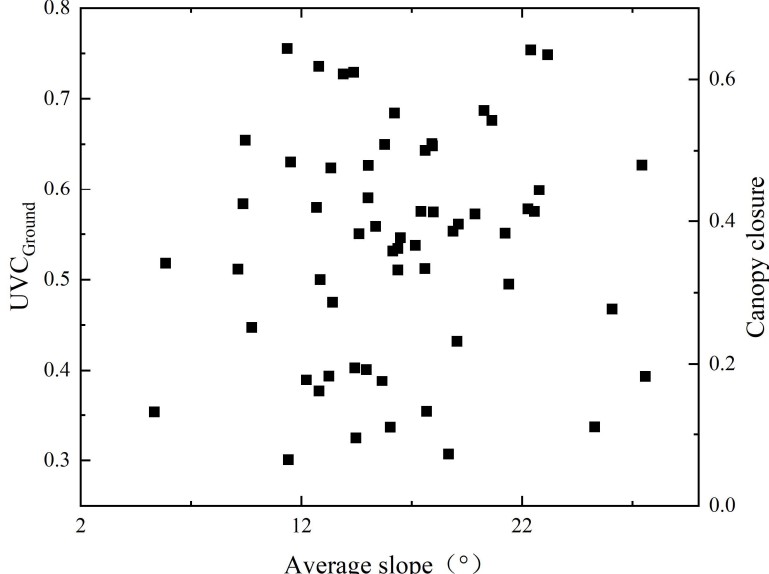

**Figure 4.** Canopy closure and topographic distribution of the sample plot. Each point in the figure corresponds to the measured information of the sample area in terms of the closures, slopes and understory vegetation.

**Table 1.** Statistical results of sample site factor information.

|  | UVC$_{Ground}$ | Average Slope (°) | Canopy Closure |
|---|---|---|---|
| mean | 0.539 | 16.466 | 0.36 |
| min | 0.301 | 5.341 | 0.1 |
| max | 0.755 | 27.596 | 0.6 |
| std | 0.124 | 4.791 | 0.119 |

UVC$_{Ground}$ refers to the measured value of understory vegetation cover. The above graph shows that the values were mainly distributed between 0.3 and 0.8, with the maximum value of 0.748 and the minimum value of 0.301. From the measured understory vegetation cover data, it is apparent that the distribution of understory vegetation in the sample sites of this study was diverse. The average slope of the sample sites was concentrated in the range of 5°–27°, which indicated that the topography of the sample sites had greater variability and the overall distribution was more reasonable.

According to the relevant definition, combined with the results of field research [3,46], the canopy closure was divided into low level (0.1–0.4) and high level (0.4–0.7) in this study, and the results of typical sample plots are shown in Table 2.

**Table 2.** Classification canopy closure results of typical sample plots.

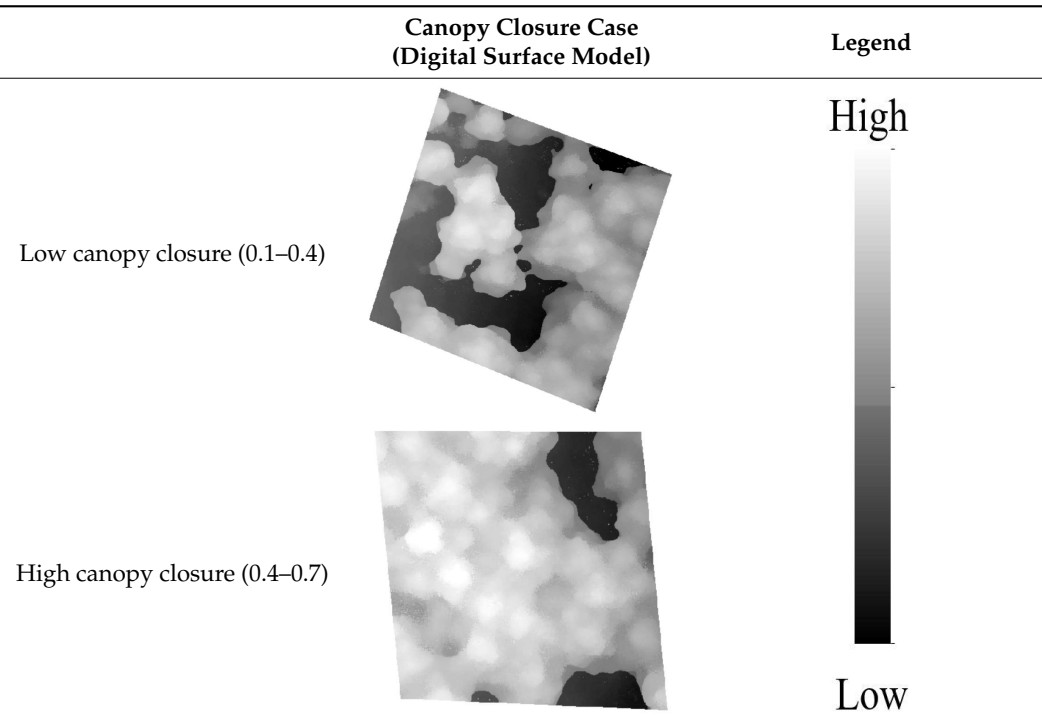

| | Canopy Closure Case (Digital Surface Model) | Legend |
|---|---|---|
| Low canopy closure (0.1–0.4) | | High |
| High canopy closure (0.4–0.7) | | Low |

Meanwhile, the study also classified the topographic relief of the sample sites into gentle slopes (6°–12°), inclined slopes (13°–22°), and steep slopes (>22°), and the results of the typical sample site classification are shown in Table 3.

**Table 3.** Classification slope results of typical sample plots.

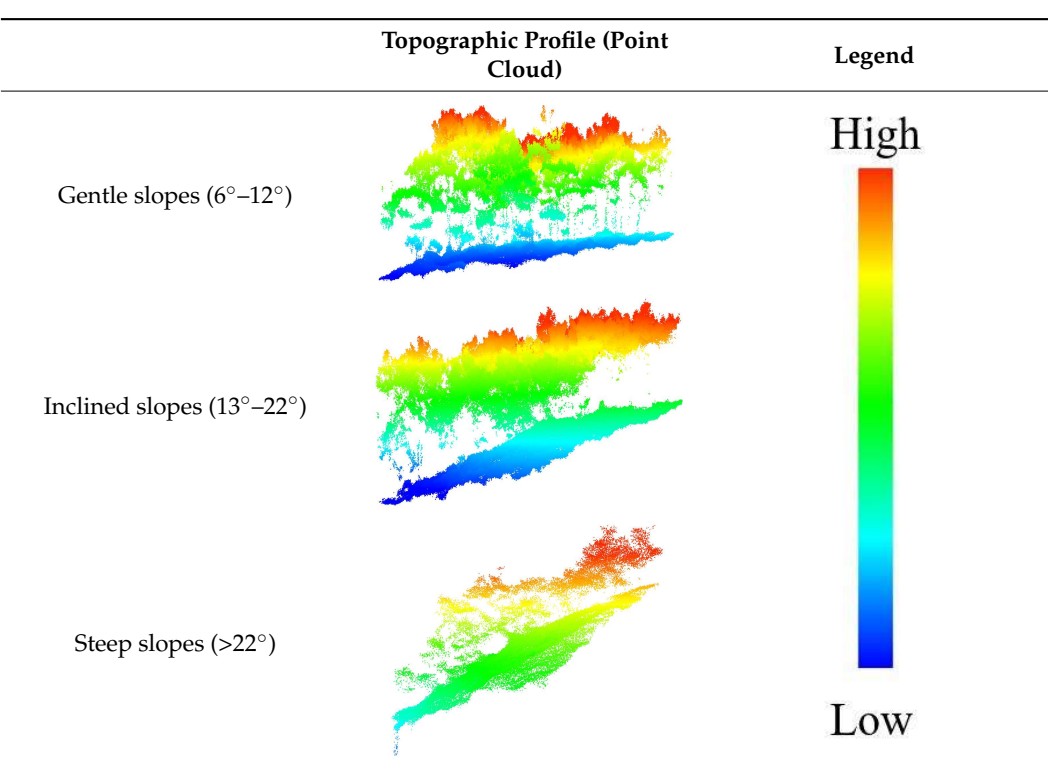

| | Topographic Profile (Point Cloud) | Legend |
|---|---|---|
| Gentle slopes (6°–12°) | | High |
| Inclined slopes (13°–22°) | | |
| Steep slopes (>22°) | | Low |

The spatial and numerical gradient distribution of topographic factors and densities in the sample plots in this study was reasonable. Therefore, it can be concluded that the sample plots set up in this study can represent the study area well and provide good sample conditions and data support for the subsequent quantification of understory vegetation cover.

### 4.2. Three-Dimensional Structural Decomposition of Pinus massoniana Forest

A feature dataset was constructed for the study area, the sampling points were selected by the farthest sampling strategy, and the point cloud semantic segmentation model was trained using the Point CNN learning model. The collected point clouds were segmented semantically. In this study, the point cloud data were divided into three categories, namely: forest canopy, understory vegetation, and ground. The farthest sampling strategy is shown in Figure 5, and the classification results are shown in Figure 6. The red dots in Figure 5 represent the seed points.

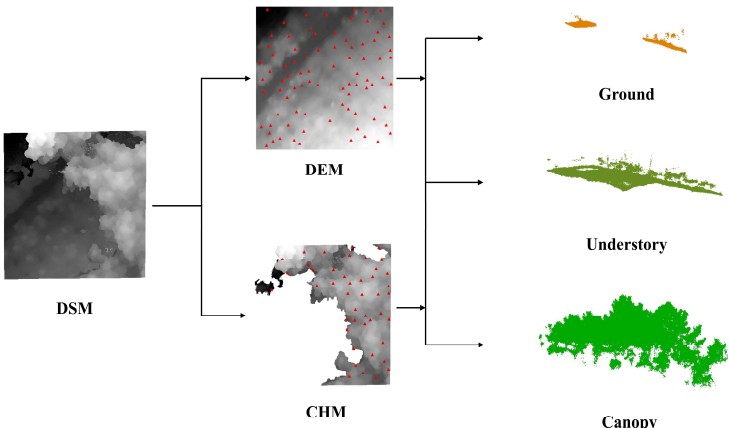

**Figure 5.** Schematic diagram of farthest sampling point strategy.

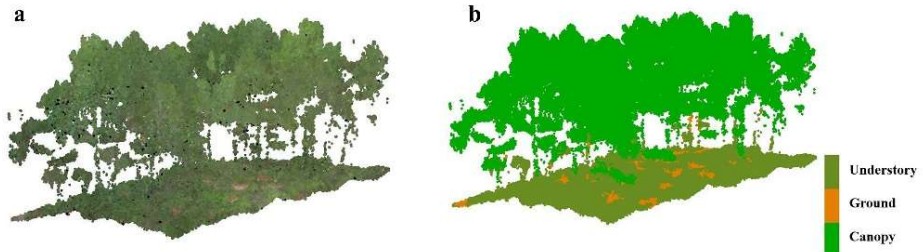

**Figure 6.** Three-dimensional structure decomposition of *Pinus massoniana* forest. (**a**): the RGB point cloud of *Pinus massoniana* forest samples; (**b**): the point cloud of *Pinus massoniana* forest 3D structure separation.

A test set for the model was constructed by selecting 70% of the data from each category in the training dataset and the remaining 30% of the data to evaluate the reliability of the training model and the realism of the prediction results. The test set samples were selected to evaluate the accuracy of the segmentation model, and the evaluation indexes included the accuracy, recall, and F1 index. To evaluate the accuracy difference between semantic segmentation and existing methods, the study classified *Pinus massoniana* forests using a point cloud cloth simulation filter algorithm based on height thresholding and obtained its classification accuracy. The accuracy evaluation results of the semantic segmentation method and cloth simulation filtering algorithm are shown in Table 4.

**Table 4.** Segmentation accuracy evaluation results. In the table, "❀/❀/❀" means "precision/recall/F1 index".

| Methods | Canopy Layer | Understory Vegetation | Ground | Overall Accuracy |
|---|---|---|---|---|
| Point CNN | 80.5/86.9/82.2 | 72.1/77.8/75.0 | 69.4/73.3/71.9 | 76.2 |
| cloth simulation filter | 78.5/82.9/80.7 | 49.8/51.4/50.6 | 29.6/44.9/37.8 | 56.4 |

Analyzing the accuracy of both results, we can see that the segmentation accuracy of the point CNN model was better than that of the cloth simulation filter algorithm for each stand feature, but for the forest canopy layer, there was little difference between the extraction accuracy of the two methods. Meanwhile, the point CNN model was more effective than the cloth simulation filter algorithm in separating the understory vegetation from the ground because the point CNN model learned and extracted the point cloud features to achieve segmentation, while the cloth simulation filter algorithm mainly extracted the height features, which was less effective in separating the features with similar elevation, such as separating the understory vegetation from the ground points.

### 4.3. Combined Active and Passive Remote Sensing to Quantify Understory Vegetation Cover

The enhanced understory vegetation information was reverse projected back to the orthophoto and voxelized to the same resolution as the orthophoto. The image was then binarized so that the understory vegetation and background were distinguishable. The calculation of the understory vegetation cover is shown in Figure 7.

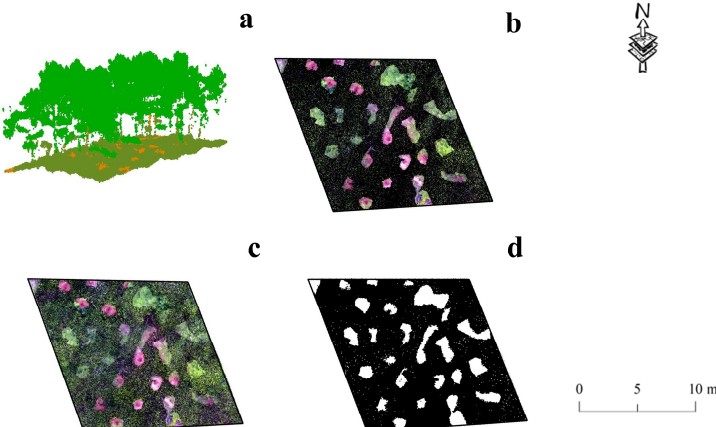

**Figure 7.** Calculation of understory vegetation cover. (**a**): The result of the decomposition of the three-dimensional structure of the sample site; (**b**): The result of the reverse projection of the understory point cloud collection back to the orthophoto; (**c**): The two-dimensional image obtained after voxelization in (**b**). In (**c**), the green part is the vegetation part, the black points are the understory vegetation points, and the pink area is the ground point. (**d**): The result of the mask after binarization of the image. The black area represents the plant area, while the white area represents the bare land area.

In order to understand the influence of canopy closure and slope on remote sensing estimation of understory vegetation coverage, the accuracy for each category was compared. In the present study, canopy closure was divided into two categories, namely low canopy cover (0.1–0.4) and high canopy cover (0.4–0.7). The slopes were divided into three categories, namely gentle slope, inclined slope, and steep slope. The results under different canopy closures are shown in Figure 8. The pink area in the figure represents the 95% confidence region of the values. The estimation accuracy is expressed by the coefficient of determination $R^2$ and the root mean square error RMSE [47]. The results are also shown in Figure 8.

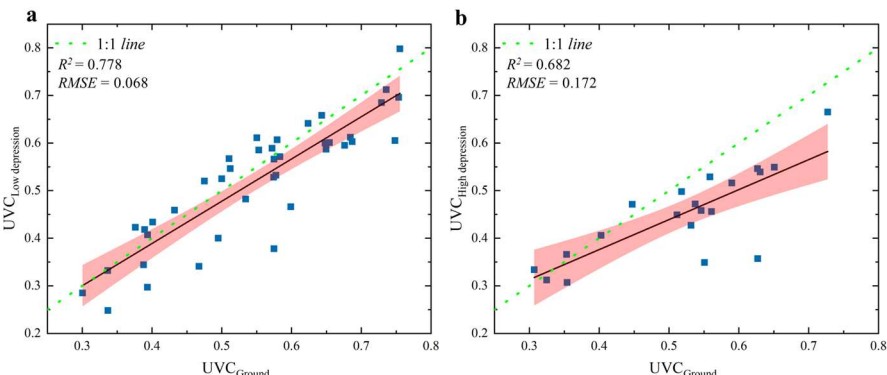

**Figure 8.** Linear regression results under different canopy closure. (**a**) Low forest densities; (**b**): High forest densities.

In parallel, the slope was divided into three categories, namely gentle slope (6°–12°), inclined slope (13°–22°), and steep slope (>22°). The results for different slopes are shown in Figure 9.

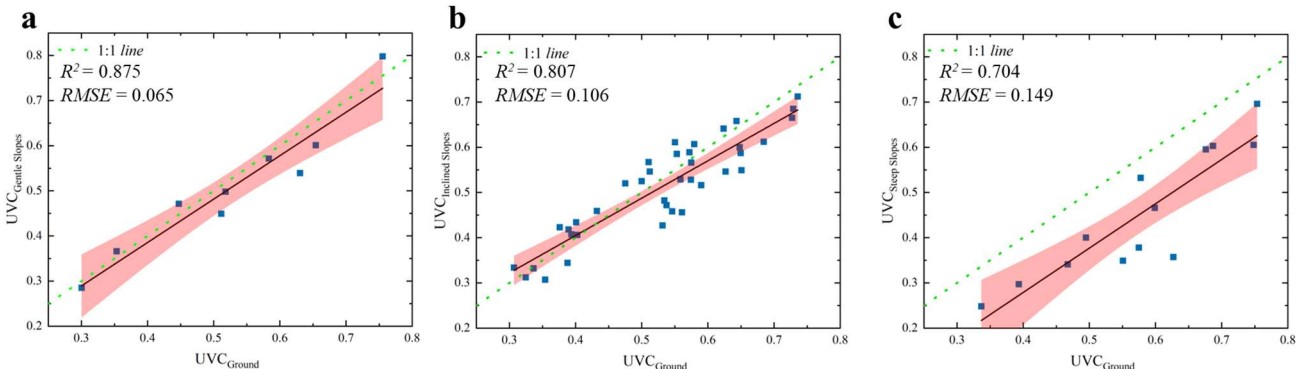

**Figure 9.** Linear regression results under different slopes. (**a**): Gentle slope conditions; (**b**): Inclined slope conditions; (**c**): Steep slope conditions.

From the above graphs, it could be seen that for the canopy closure, the estimation accuracy under low values was higher than that under high values; for the slope, the estimation of understory vegetation cover under gentle slope conditions was the most accurate, followed by slope conditions, and the accuracy under steep slope conditions was the lowest.

## 5. Discussion

### 5.1. Effect of Point Cloud Segmentation Methods on Quantifying Understory Vegetation Cover

The results of the separation of the three-dimensional structure of the *Pinus massoniana* forest stand to determine the accuracy of subsequent remote sensing to quantify the understory vegetation cover. In this study, the point CNN model was used to separate the point clouds of the *Pinus massoniana* forest samples, and its overall segmentation accuracy was 74%. The overall segmentation accuracy was 55.7% when using the traditional point cloud filtering algorithm. The results show that the segmentation accuracy of the sample point clouds using the Point CNN model was greater than that using the point cloud cloth simulation filter. However, comparing the results of the Point CNN model in an urban context [48], the segmentation accuracy of Point CNN for forest scenes is worse than that for urban features.

The point clouds of different features within urban scenes varied greatly, unlike in urban scenes, where the point cloud information within the forest, whether in true color or internal spectral features, had much less feature variation. Coupled with the higher

flight altitude of airborne LiDAR in the general study of urban-related research, this led to a reasonable interval of point cloud density, which was unlikely to lead to point cloud information redundancy or gradient explosion in the model training, resulting in a higher accuracy of the model in urban contexts. However, in the forest scene, the vegetation pattern was more irregular, and the point cloud density in the sample area had to be increased to make the point cloud fit the features better. Therefore, the segmentation accuracy is reduced. The overall accuracy of the method was still higher than that of the point cloud filtering method.

The point cloud cloth filtering algorithm is strongly dependent on the relative elevation of the point cloud simulation algorithm [49]. In certain plain areas, the method extracted the forest topography with high accuracy. However, this study centered on a hilly region in southern China, and the dramatic changes in point clouds generated by topographic undulations had a more serious impact on the relative elevation of the point clouds in each part of the forest stand, resulting in fair results for the segmentation of the forest canopy layer far from the ground. As the height difference between the understory vegetation and the ground was small, these feature points were easily confused, which led to poor results when separating the understory vegetation from the ground.

### 5.2. Influence of Point Cloud Inverse Projection Algorithm and Slope on Quantifying Understory Vegetation Cover

To quantify the understory vegetation cover using combined active and passive remote sensing, the point cloud was mapped back to two dimensions using the point cloud inversion projection algorithm. The mapped projection surface had an angle with the ground; the larger the slope, the larger the angle between the ground and projection surface. In a similar study, a point cloud back-projection algorithm was used to estimate understory vegetation cover in the Sehanba area of Hebei Province [10]. Due to the homogeneity of the Seyhanba area, the terrain was gentle, and the angle between the slope and the projection surface was not large; therefore, the study results were less affected by the slope. In this study, from the results under different conditions, it could be seen that the accuracy of quantifying understory vegetation cover was not much different from that of similar studies, even for high canopy closure and steep-slope samples with the lowest accuracy.

The sample site for this study was in a hilly region in southern China. The region was characterized by an undulating terrain and soil erosion. Photos were taken vertically on the ground, while data collected by the UAV systems had an angle between the ground and the projection surface. The greater the slope, the greater the difference between the two levels. Therefore, when using the point cloud back-projection algorithm to quantify the understory vegetation information of steep slope sample plots, the quantification accuracy was lower than that of gentle slope sample plots.

### 5.3. The Applicability of Quantitative Understory Vegetation Methods

The study area is in Hetian Town, Changting County, Fujian Province, a place where soil erosion is very serious in southern China. Although it has been treated and improved for many years, its forest structure is still relatively simple, and most natural forests contain only trees and lower herbaceous plants, while shrubs rarely appear or are mostly dead; therefore, only herbaceous vegetation was considered as understory vegetation in this study. The method adopted in this study to decompose the forest structure and enhance the understory vegetation information is the semantic segmentation of the forest point cloud, which relies on a segmentation dataset trained using a deep learning model. In this study, the forest structure was divided into two layers, trees and grasses, when training the dataset. In the face of a healthier and more complex forest ecological structure (i.e., trees, shrubs, grass, and mosses), this method is also applicable to forest structure decomposition if the training segmentation dataset contains shrubs or mosses [50].

However, the cost of using the deep learning algorithms used in this study was significant relative to traditional machine learning methods that can be used in forested

situations. In order to obtain proper results, a highly configured graphics card with high graphics memory and almost two days of uninterrupted training were required to obtain a constructed analysis result. Furthermore, the cost of acquiring point clouds using airborne LiDAR was also greater than that of acquiring point clouds using multispectral drones. Additionally, it is currently not possible to use relatively simple UAV point clouds for the training and decomposition of forest 3D structures.

*5.4. Effect of Canopy Closure on Quantifying Understory Vegetation Cover*

The canopy cover can affect the ability of acquiring understory vegetation information by the UAV LiDAR system. Since the laser spot of the Zenith L1 lens used in this study was an elongated ellipse, the laser could not penetrate the diameter of the forest gap smaller than the long axis of the spot, and thus affected the acquisition of understory vegetation information in the forest interior under this gap. In this study, the average density of understory vegetation information points was 600 to 800 points/m$^2$ in the low-density sample sites, while the average density of understory information points was 400 to 600 points/m$^2$ in the high-density sample sites. The abundance of understory information was significantly lower in the high-density sample sites than that in the low-density sample sites. Therefore, the estimation accuracy under high canopy closure was lower than that under low canopy closure.

## 6. Conclusions

Based on the UAV LiDAR point cloud data and UAV orthophoto, this study trained a *Pinus massoniana* forest segmentation dataset and separated the three-dimensional structure of the *Pinus massoniana* forest using the point CNN model, and two-dimensional mapping of the three-dimensional information of the understory vegetation was carried out using the point cloud inverse projection technique. The two-dimensional information voxels of the understory were then binarized to estimate the understory vegetation cover. The accuracies under different conditions were also evaluated. The results show that the point cloud semantic segmentation method based on the point CNN model was accurate in separating the three-dimensional structure of the forest scene, and the accuracy of the method met the requirements for separating the understory vegetation. Meanwhile, the joint active–passive remote sensing quantitative understory vegetation method proposed in this study could accurately evaluate the understory vegetation at the sample site under different factor conditions, which provided a theoretical basis and technical support for the quantitative estimation of understory vegetation.

Only one flight altitude was performed at each sample site to acquire point cloud information. In a subsequent study, data from other flight altitudes of the sample sites will be collected to ensure the diversity of the study data in terms of point cloud density, in order to explore the optimal segmentation density of point clouds using deep learning models. Meanwhile, to address the problem that the single-angle LiDAR point cloud was obscured by the forest canopy, a follow-up study will consider the use of multi-angle LiDAR to collect the sample land data to explore the forest interior.

**Author Contributions:** Conceptualization, F.W.; Data curation, R.W., S.T. and L.S.; Formal analysis, F.W.; Funding acquisition, J.L. and F.W.; Investigation, R.W., and S.Z.; Methodology, R.W.; Project administration, J.L. and K.Y.; Resources, R.W.; Software, R.W.; Supervision, J.L. and K.Y.; Validation, T.B. and S.T.; Visualization, R.W. and L.S.; Writing—original draft, R.W. and F.W.; Writing—review & editing, T.B. and F.W. All authors have read and agreed to the published version of the manuscript.

**Funding:** This research was funded by The National Science Fund for Young Scholars, grant number (41901387), and Fujian Provincial Natural Science Foundation, grant number (2022J05031).

**Institutional Review Board Statement:** Not applicable.

**Informed Consent Statement:** Not applicable.

**Data Availability Statement:** Not applicable.

**Acknowledgments:** This research was supported by the Forestry College of Fujian Agriculture and Forestry University. We are grateful to DJI Fujian Branch for teaching drone flight technology. We also appreciate Chunwang Cai for helping with the field investigation.

**Conflicts of Interest:** The authors declare no conflict of interest.

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
