# Peer review of "Quantifying Understory Vegetation Cover of Pinus massoniana Forest in Hilly Region of South China by Combined Near-Ground Active and Passive Remote Sensing"

_drones, doi:10.3390/drones6090240_

Round 1

Reviewer 1 Report

The authors proposed a method which combined with the UAV orthophoto and airborne LiDAR data to detect the understory vegetation. The discussion is convincing and the conclusions sound reliable. In general, the paper is well organized and convincing.

1. The shortcomings of the current study in the Abstract could be added.

2. The version information of the ArcGIS software should be added.

3. Whats limitations for the proposed method in practical application? If possible, please give some examples.

The following paper could be referenced.

Hepi H. Handayani, Arizal Bawasir, Agung B. Cahyono, Teguh Hariyanto & Husnul Hidayat (2022) Surface drainage features identification using LiDAR DEM smoothing in agriculture area: a study case of Kebumen Regency, Indonesia, International Journal of Image and Data Fusion, DOI: 10.1080/19479832.2022.2076160

Author Response

Response to Reviewer 1 Comments

Point 1. The shortcomings of the current study in the ‘Abstract’ could be added.

Dear Reviewer: Through a review and summary of existing research control of the abstract structure, we added the following sentence after the second sentence of the abstract:" Nevertheless, the multi-source data synergy mechanism has not been fully revealed in the remote sensing quantification of understory vegetation cover due to the insufficient match between the point cloud 3D data obtained from active and passive remote sensing systems and the UAV orthophoto resulting in the abundance of understory vegetation information not being represented in two dimensions" as the shortcomings of the current study.

Point 2. The version information of the ArcGIS software should be added.

Dear Reviewer: In this paper, the software version for point cloud semantic segmentation using ArcGIS software was ArcGIS pro2.9, and the software version for mapping, spatial interpolation and geostatistical analysis using ArcGIS software was ArcMap in ArcGIS 10.8. ArcMap 10.8 has built-in functions for mapping, spatial interpolation and geostatistical analysis that were not too buggy, so this paper uses both versions of ArcGIS software to operate. The relevant information was indicated in the original article.

Point 3. What’s limitations for the proposed method in practical application? If possible, please give some examples.

Dear Reviewer: In fact, in practical application, the method proposed in this thesis for estimating understorey vegetation cover has certain limitations. Firstly, due to hardware issues, the laser spot of the Zenith L1 lens used in this study was a thin ellipse, resulting in the laser not being able to penetrate a forest gap diameter smaller than the long axis of the spot, thus affecting the acquisition of information on the understorey vegetation within the forest under that gap. Therefore, this study was likely to be limited to sample plots with large gaps or small closures. Secondly, in the forest scenes, where the vegetation pattern was irregular, it was necessary to increase the point cloud closure within the sample plot in order to make the point cloud fit the features better. However, this will lead to redundant point cloud information, which is more likely to cause gradient explosion in the model training, resulting in poor accuracy of the model. Thirdly, the cost of using the instruments used in this experiment was high. The DJI M300 UAS sells for approximately US$20,000 and the Zenith L1 lens and P1 lens sell for a combined total of approximately US$13,000. The battery and charging system together cost about US$3,000. What's more, all the instruments together weigh over 30kg, making them difficult to carry. The batteries take too long to charge and drain too quickly, making them costly in terms of time spent in use. Finally, the cost of using deep learning algorithms was also greater. A high configuration graphics card with high video memory and almost a day of non-stop training was required to get the appropriate results.

The following paper could be referenced.

Hepi H. Handayani, Arizal Bawasir, Agung B. Cahyono, Teguh Hariyanto & Husnul Hidayat (2022) Surface drainage features identification using LiDAR DEM smoothing in agriculture area: a study case of Kebumen Regency, Indonesia, International Journal of Image and Data Fusion, DOI: 10.1080/19479832.2022.2076160

Dear Reviewer: The literature has been seen by the author and has been cited in this thesis.

Reviewer 2 Report

Comments to authors:

Understory vegetation cover in forests is an interesting and meaningful topic. The authors investigated a method of combining UAV orthophoto and LiDAR data to detect the understory vegetation. Although it is a good topic, I still have some concerns and don’t think it is good enough to be accepted right now. The paper may be accepted after following comments are seriously considered.

Major and minor revisions:

1.     The definition of understory vegetation is not clear enough. According to the current definition in this study, the understory vegetation is only herbaceous vegetation, no shrub, right? Is it because the area belongs to southern China or something else? So, this method in this study could be used for one-layer forest (trees and grass)? Please clarify the definition in the introduction and applicability in the discussion.

2.     For section 3.1, please state the purpose of the photographs first and then describe the details of them. Otherwise, the readers may wonder why the photographs are used for. Line 136-137, “The resolution of the photo is 4 000 136 x 6 000 pixels.” Do you mean the width of photograph? Please revise or clarify it.

3.     For figure 2, it is not the flow chart of the entire experiment. Figure 2 should include all the steps and logic of the methodology.

4.     For the data pre-processing, LiDAR360 classified the point cloud into forest canopy, understory vegetation and ground points. What is the difference of the understory vegetation in LiDAR360 and that the authors want to classified in this study? Please clarify it.

5.     Line 220-221, do you mean that PGreen represents the pixels occupied by the understory vegetation in the photo? It is not green vegetation. Please consider using Punder or other notations. So, UVC is calculated for each photo, right?

6.     The logic of methodolgy is not clear enough. The titles of section 3.2.4 and 3.2.5 do not make sense. Please revise them. I don’t think it is necessary to show the equation of Kriging interpolation. Please make the logic of the method match the new figure 2.

7.     What is the purpose of section 4.1 and figure 4? If it only described the field measurement, it does not belong to the result section. Please move it to the data section. Please use a table to show the statistics of measured data instead of figure 4.

8.     Do you have any reference or rules to support the classification of slopes? The slope of 13-22 is only called “slope”?? For table 2, please add the range for each slope types. In second column, please use the same view to show the three slopes. The difference between 6-12°and 13-22°is little in the figure.

9.     Similar to the previous comment, any reference or reason to support the classification of canopy closure (0.1-0.4, and 0.45-0.7)? What about 0.4-0.45?

10.  Another concern is the conclusion about the effect of point cloud inverse projection algorithm and slope on quantifying understory vegetation cover. There is no statistical comparison. The conclusion is unsound. Why don’t you conduct a two-way ANOVA (i.e., algorithm and slope are two factors) to testify it?

Line 320-321 on page 10, there is no statistical support.  

11.  I don’t think table 4 is useful. The authors could show the results in the figure 8 and 9. The regression equation is not necessary. Please reorganize the results.

12.  Most figures in the manuscript should be improved.

(1)  Figure 1 is not clear enough. The points representing the sample plots are too big. Please revise it.

(2)  Figure 2 should be improved as the comment above.

(3)  Please move the description of notation from the text to the figure as a note. For example, line 257 on page 8, move the UVCground explanation as a note of figure 4.

(4)  Figure 6, please explain (a) and (b) in the figure caption. Why are there some words in other language? Please prepare the manuscript carefully.

(5)  For figure 7, please explain (a) to (d) in the figure caption, and move the explanation of pink area in the note or using legend.

(6)  Please reorganize the figure 8 and 9 as the comment above and explain (a) and (b) in the figure caption.

13.   The authors mentioned “soil erosion” several times in this study. Is the prediction of understory vegetation cover only meaningful to soil erosion? Please clarify the meaning of this methodology or this study to remote sensing field. Could we use this method for other areas?

14.   Some expression in this paper is confused or redundant. Please revise them. Here are a few examples:

(1)  Line 28 on page 1, it should be “southern hilly area of China”. You didn’t test it in other places.

(2)  Line 35 on page 1, “hilly soil erosion area”? How to define it? Please revise it.

(3)  Line 111 on page 3, do you mean “in hilly areas of southern China”?

(4)  Line 200 on page 6, do you mean “two-dimensional visualization”?

(5)  Please revise the title of section 4.2.

(6)  Line 312 and 314 on page 10, why use “cloth simulation filter algorithm” on line 312, but “fabric filtering algorithm” on line 314? Are they two algorithms?

(7)  Line 315 on page 10, it should be a note in table 3. 

(8)  Line 365-366 on page 12, it should be moved to discussion secction.

Revisions needed: There are other mistakes and confusions. I cannot list them all. The manuscript should be polished by a native speaker.

Author Response

Response to Reviewer 2 Comments

Point 1.     The definition of understory vegetation is not clear enough. According to the current definition in this study, the understory vegetation is only herbaceous vegetation, no shrub, right? Is it because the area belongs to southern China or something else? So, this method in this study could be used for one-layer forest (trees and grass)? Please clarify the definition in the introduction and applicability in the discussion.

Dear Reviewer: The common definition of understory vegetation is the collection of all vegetation, including shrubs, herbs and mosses, with distinct stratification and spatial heterogeneity under the tree canopy in a forest system. According to the current definition in this study, the understory vegetation was defined as herbaceous vegetation only, without shrubs, because the study area of this paper was selected in Changting County, Fujian Province, a hilly soil erosion area in southern China. Due to years of soil erosion in this area, the soil was deprived of organic matter, and despite years of treatment, the forest structure was still relatively simple, with most natural forests containing only upper layer trees and lower layer herbs, while shrubs rarely appeared or were dead. Therefore, only herbaceous vegetation was considered as understory vegetation in this study. The method for decomposing the forest structure in this paper relies on the segmented dataset trained by the deep learning model being used and the dataset trained in this study can be used to decompose a single layer of forest (trees and grass). If the training segmentation dataset contains shrub or moss samples, then the method of this study is also applicable to more complex forest structure decomposition (treesshrubsgrassmosses).

The definitions in the introduction and the applicability-related parts of the discussion have been clarified. The relevant modifications are in part as follows:

Add in Line38-40:“including shrubs, herbaceous vegetation and mossy vegetation”;

Add in Line482-500

“The applicability of quantitative understory vegetation methods

The study area is in Hetian Town, Changting County, Fujian Province, a place where soil erosion is very serious in southern China. Although it has been treated and improved for many years, its forest structure is still relatively simple, and most natural forests contain only trees and lower herbaceous plants, while shrubs rarely appear or are mostly dead; therefore, only herbaceous vegetation was considered as understory vegetation in this study. The method adopted in this study to decompose the forest structure and enhance the understory vegetation information is the semantic segmen-tation of the forest point cloud, which relies on a segmentation dataset trained using a deep learning model. In this study, the forest structure was divided into two layers, trees and grasses, when training the dataset. In the face of a healthier and more complex forest ecological structure (i.e., trees, shrubs, grass, and mosses), this method is also applicable to forest structure decomposition if the training segmentation dataset con-tains shrubs or mosses[47].

However, the cost of using the deep learning algorithm used in this study was greater. In order to obtain proper results, a highly configured graphics card with high graphics memory and almost two days of uninterrupted training were required to ob-tain a constructed analysis..”

Point 2.     For section 3.1, please state the purpose of the photographs first and then describe the details of them. Otherwise, the readers may wonder why the photographs are used for. Line 136-137, “The resolution of the photo is 4 000 x 6 000 pixels.” Do you mean the width of photograph? Please revise or clarify it.

Dear Reviewer: Photographs taken by the camera were used to calculate the actual measurement of understory vegetation cover.

UAV airborne LiDAR data were used to probe the three-dimensional structure of the forest. In forest areas, there  were situations where the upper tree branches and leaves obscure the understory vegetation, which has a large accuracy error in detecting the three-dimensional forest structure and quantifying the understory vegetation. The laser pulse signal emitted by airborne LiDAR technology has certain penetration ability to vegetation, which can largely reduce the information loss caused by vegetation branch and leaf shading, and thus obtain the real 3D information in the forest sample area. UAV visible photos  were used to stitch together UAV orthophotos. Since the understory vegetation cover is a two-dimensional index and the point cloud is a three-dimensional information, the point cloud of understory vegetation characterized by airborne LiDAR needs to be two-dimensionalized with the orthophoto as the carrier, and the three-dimensional information can be two-dimensionalized before the subsequent calculation of understory vegetation cover.

Line 136-137, “The resolution of the photo is 4 000 x 6 000 pixels.”

This sentence means that each picture is 6,000 pixels long and 4,000 pixels wide; I may not have been accurate in describing the size of the shot in this way, and have changed it to 6,000 pixels long and 4,000 pixels wide per picture. The schematic diagram is as follows.

In view of the above,the corresponding changes are as follows:

Add in Line142-143:“which were used to calculate the actual measurement of understory vegetation cover.”;

Add in Line149-150:“These UAV visible light photos were used to stitch the UAV orthophotos.

Add in Line161-162:“UAV airborne LiDAR data were used to probe the three-dimensional structure of the forest

Modify in Line143-144:“Each image was 6,000 pixels in length and 4,000 pixels in width.

Point 3.     For figure 2, it is not the flow chart of the entire experiment. Figure 2 should include all the steps and logic of the methodology.

Dear Reviewer: The original diagram mainly focuses on the combined active-passive remote sensing process, which is indeed not a complete technical flow chart. Therefore, the part of actual ground measurements and the part of joint active-passive remote sensing to quantify the understory vegetation and analyze it were added to the original diagram. The modified results are as follows.

Related graphics have been modified in the text.

Point 4.     For the data pre-processing, LiDAR360 classified the point cloud into forest canopy, understory vegetation and ground points. What is the difference of the understory vegetation in LiDAR360 and that the authors want to classified in this study? Please clarify it.

Dear Reviewer: In this study, we first trained the point cloud segmentation dataset using the built-in machine learning algorithm of LiDAR360, so as to classify the forest 3D structure point clouds into three categories: forest canopy, understory vegetation and ground points. However, using LiDAR360 machine learning segmentation method to segment the forest stand 3D point cloud, the results of separating the forest canopy and understory parts have high accuracy, while the results of understory vegetation and ground points have poor accuracy and need to be further processed. In this paper, we use the classification results for further modification and make a semantic segmentation dataset based on the results. The main reason why this paper uses LiDAR360 machine learning algorithm to segment 3D point clouds as a data pre-processing session was to reduce the workload of producing semantic segmentation datasets, while effectively retaining point cloud coordinates and other ancillary information to ensure that there was no data loss in them.

The comparison of LiDAR360 segmentation 3D point cloud results and semantic segmentation results is shown below.

LiDAR360 segmentation results

semantic segmentation results

Therefore,the corresponding changes are as follows:

Modify in Line195-203”The LiDAR data were processed using LiDAR360 (https://www.lidar360.com/) software. The acquired raw LiDAR point cloud data were pre-processed through crop-ping and de-noising, and the built-in machine learning algorithm of the software was used to pre-classify the point cloud into three categories, namely: forest canopy, un-derstory vegetation and ground points. The machine learning classification tool of the software uses random forests to classify the point cloud data. By manually editing the categories of typical data in the same batch, the model was trained and subsequently batched to process a large amount of data. The classification results were processed further to construct a semantic segmentation dataset.”

Point 5.     Line 220-221, do you mean that PGreen represents the pixels occupied by the understory vegetation in the photo? It is not green vegetation. Please consider using Punder or other notations. So, UVC is calculated for each photo, right?

Dear Reviewer: Lines 220-221, PGreen in equation (2) represents the pixels occupied by the green image element in the photo. The photo of the understory vegetation taken in this paper was taken at a height of about 1.7m from the ground, and the photo contains only two elements: green vegetation and the ground, according to the height reference, the green vegetation in the photo can be considered as the understory vegetation, thus the original paper used PGreen to represent the pixels occupied by the green pixels in the photo. However, such a reference may cause misunderstanding, so, after modification, this paper considers using PUndVeg to refer to the pixels occupied by green pixels in the photo.

Referring to the new technical flowchart, the actual UVC ground measurements are the average of the thresholds calculated for each photo of the sample plot, as shown in the flowchart.

Modify in Line244-250:“PGreen”→“PUndVeg

Point 6.     The logic of methodolgy is not clear enough. The titles of section 3.2.4 and 3.2.5 do not make sense. Please revise them. I don’t think it is necessary to show the equation of Kriging interpolation. Please make the logic of the method match the new figure 2.

Dear Reviewer: Referring to the new technical flow chart, the methodology section was updated as follows:

First, the UAV visible photos and airborne LiDAR data of the Pinus massoniana forest sample sites were acquired, and all remote sensing data were confirmed to be within the WGS_1984_UTM_Zone_50N geographic coordinate frame. Second, the UAV visible-light photos were used to stitch the UAV orthophoto images, and the point cloud information was obtained using airborne LiDAR data. Third, the point cloud data were pre-processed, and a semantic segmentation model was constructed for the Pinus mas-soniana forest. This was done based on which the three-dimensional structure of the forest stand was decomposed with high precision and the three-dimensional infor-mation of the understory vegetation was enhanced. Subsequently, the enhanced un-derstory information was back-projected to the UAV orthophoto using the point cloud back-projection algorithm. Thereafter, the aggregated data were voxelized and binarized. The understory vegetation was quantified using binarized data. Meanwhile, based on the ground and canopy point sets obtained using semantic segmentation, the slope and canopy closure of the statistical sample sites were extracted and used as underlying factors. The near-ground photographs of the sample plots were taken, and the vegeta-tion patterns were outlined using a threshold algorithm combined with a spatial in-terpolation algorithm to obtain the ground truth values of the understory vegetation. Finally, an accuracy analysis of the remote sensing estimation of understory vegetation under different slope and canopy closure conditions was performed. The experimental procedure is shown in Figure 2.The titles of Sections 3.2.4 and 3.2.5 have been modified in accordance with the Methodology content. The corresponding changes are as follows

3.2.4 Method of calculating the ground-truthing value of understory vegetation cover

3.2.5 Calculation of slope and canopy closure

In 3.2.5, the section on kriging interpolation has been removed

Point 7.     What is the purpose of section 4.1 and figure 4? If it only described the field measurement, it does not belong to the result section. Please move it to the data section. Please use a table to show the statistics of measured data instead of figure 4.

Dear Reviewer: Section 4.1 of this paper is about the process of calculating the ground-truthing value of understory vegetation cover according to 3.2.4 Method of calculating the ground-truthing value of understory vegetation cover and 3.2.5 Method of Sample Site Information Statistics, the process of obtaining the ground-truthing value of understory vegetation cover was briefly described. It also described the process of calculating the slope information and lushness information of the sample site after obtaining the ground point set and canopy point set of the sample site and the related results.

Figure 4 is a schematic representation of the average slope information and forest depression information calculated for all the sample plots in this study, as well as the ground truth values of the understory vegetation. As can be seen from the graph, the horizontal axis of the graph is the average slope of the sample plots, the left vertical axis indicates the ground truth value, and the right vertical axis indicates the depression of the sample plots. The figure indicates that the sample plots selected in this paper conform to the naturally distributed.

The information expressed in Figure 4 has been transferred to Table 1. It should be clarified that, unlike vegetation cover, forest depression was expressed as a multiple of 0.1 according to the relevant Chinese forestry standards. However, to make the data division more accurate, the canopy closures  were saved in two bits and  were rounded to 0 or 5 in this paper. The information on the slope of the sample site was kept to three decimal places.

The relevant content has been modified in the original article.

Add in Line278-284:“It should be clarified that, unlike vegetation cover, the value of canopy closure was ex-pressed as a multiple of 0.1 according to the relevant Chinese forestry standards. However, to make the data division more accurate, the value of canopy closure was kept at two decimals and rounded to the nearest half or whole number in this study. The slope of the sample site was maintained at three decimals

Point 8.     Do you have any reference or rules to support the classification of slopes? The slope of 13-22 is only called “slope”?? For table 2, please add the range for each slope types. In second column, please use the same view to show the three slopes. The difference between 6-12°and 13-22°is little in the figure.

Dear Reviewer: The idea of dividing slope in this paper was as follows: firstly, the 30m resolution DEM of Hetian town in Changting County was obtained through the geospatial data cloud (https://www.gscloud.cn/home), and the average slope of Hetian town and the average slope distributed in the sample sites of Hetian town were calculated. The method does not consider the influence of land classification, and its calculation of the average slope of He Tian Town was about 6.4°, so the slope of this paper was calculated from 6°. According to the relevant Chinese slope classification standards, "6°-15° was a gentle slope, 16°-25° was a inclined slope, and 26°-35° was a steep slope". The standard was fine-tuned according to the actual situation: 1. There was no sample land with slope above 30° in this study; 2. Considering that the topographic changes in elevation within the sample land will be averaged, so the slope division intervals  were all relaxed by 3°, and finally the results divided in this paper  were obtained.

To avoid misunderstandingmodify the Title ofslope of 13-22from slopeto Inclined slope. Relevant parts have been modified in the full text.

Slope range has been added to the original text. And changed images and cut to the same view.

Point 9.     Similar to the previous comment, any reference or reason to support the classification of canopy closure (0.1-0.4, and 0.45-0.7)? What about 0.4-0.45?

Dear Reviewer: In traditional Chinese stand surveys, forest depression is usually classified using the concept of grading, i. e., the depression is divided into four classes: no forest, low, medium, and high, corresponding to the degree of canopy closure as follows.

Canopy closures were graded as 0 (no forest) with canopy closure (%) of [0, 20); 1 (low) with canopy closure (%) of [20, 40); 2 (medium) with canopy closure (%) of (40, 70); and 3 (high) with canopy closure (%) of [70, 100].

However, because the study area is in a place where soil erosion is serious and the main tree species is pine, most of the natural forests in this area are in a low to medium depression state. Therefore, the criteria were fine-tuned according to the actual situation: the depression level 0 and 1 were combined and collectively referred to as low depression; the medium depression level was retained and the high depression level was discarded, and the high depression level in this paper was in fact the medium depression level.

Under the calculation and classification criteria of lushness in this paper, sample sites with lushness of 0.4-0.45 do not exist. In order not to cause misunderstanding, the range of high closures was adjusted to 0.4-0.7.

The relevant content has been modified in the original article.

Point 10.  Another concern is the conclusion about the effect of point cloud inverse projection algorithm and slope on quantifying understory vegetation cover. There is no statistical comparison. The conclusion is unsound. Why don’t you conduct a two-way ANOVA (i.e., algorithm and slope are two factors) to testify it?

Line 320-321 on page 10, there is no statistical support. 

Dear Reviewer: We thank the reviewers for your comments on the discussion of the section 5.2 Slope and point cloud back projection algorithm for remote sensing quantification of understory vegetation cover. Your questioning of this section is very representative and the proposed solution is very useful. Your proposal to conduct a two-way ANOVA (i.e., algorithm and slope are two factors) to demonstrate that slope and point cloud back-projection algorithms affect remote quantification of understory vegetation cover is the best response to your query. However, the algorithm is not a specific impact factor and the author is unable to visualize the algorithm as an impact factor. Hence, in the face of your query, I will explain it in the most original way of thinking. I hope it can effectively respond to your query.

It can be seen that the above figure is a schematic diagram of how the point cloud back projection algorithm works in the mountainous study area. ∠A is the average slope of the sample plot. Since the internal error angles are equal, ∠A and ∠B are equal in degree. ∠B is also the angle between the projection surface and the ground. Therefore, the ratio of the point cloud projection result to the true value is 1/cos∠B. We divide the resultant remote sensing value in the steep slope condition by 1/cos∠B and make another linear fit, and the result is shown in figure below

Compared with the results without the cos∠B treatment, the accuracy of the treated results is significantly higher. R2 increases from 0.704 to 0.796 and RMSE decreases from 0.149 to 0.118. Therefore, under that experiment, I believe that for the paragraph on the effect of slope and algorithm on quantifying understory vegetation is valid.

Line 320-321 on page 10, there is no statistical support. 

Regarding this point, the data in Table 4 now reflect this conclusion. In Table 4, for canopy segmentation accuracy, both are about the same, and for other features, the segmentation accuracy is more different. The result is derived from a pre-experiment we performed, i.e., decomposing the forest structure using two algorithms separately and counting the error of both.

Point 11.  I don’t think table 4 is useful. The authors could show the results in the figure 8 and 9. The regression equation is not necessary. Please reorganize the results.

Dear Reviewer: This information table was prepared in this paper because in addition to the linear fit plots, we wanted to show more intuitively the accuracy and regression equations for remote sensing quantification of understory vegetation cover under each condition. This may result in redundant information. However, from the author's point of view, the information in the table may give a more intuitive indication of the accuracy information when the figure does not indicate the specific error. Yet this may result in the same information being presented in two separate ways, which can lead to a sense of information tearing for reviewers. Consequently, we added the relevant accuracy information to the plots and removed the linear fitted regression equations for each condition based on your comments. In fact, after your reminder and the statistical knowledge learned by the author, I also feel that the relevant regression equation is not very meaningful.

In view of the above, Table 4 has been removed from the paper, and the R2 and RMSE information has been noted in Figure 8 Figure 9, respectively, along with the removal of the regression equation.

Point 12.  Most figures in the manuscript should be improved.

(1)  Figure 1 is not clear enough. The points representing the sample plots are too big. Please revise it.

Dear Reviewers:The points in the figure indicating the sample sites have been adjusted downward. The original point size was 25 units, and the modified point size is 15 units. The unit size is defined by ArcGIS.

Relevant figures have been modified in the paper. The modified results are as follows

(2)  Figure 2 should be improved as the comment above.

Dear Reviewers:The part of actual ground measurements and the part of joint active-passive remote sensing to quantify the understory vegetation and analyze it  were added to the original diagram.Relevant figures have been modified in the paper. The modified results are as follows.

(3)  Please move the description of notation from the text to the figure as a note. For example, line 257 on page 8, move the UVCground explanation as a note of figure 4.

Dear Reviewers: The information related to Figure 4 to be transferred to Table 1.

Meanwhile, the meaning of UVCground has been explained in the text. Similar issues have been checked and revised.

Add in Line265-266:“UVCGround is the ground truth value of understory vegetation cover.”;

(4)  Figure 6, please explain (a) and (b) in the figure caption. Why are there some words in other language? Please prepare the manuscript carefully.

Dear Reviewers: The relevant information is indicated in the title of Figure 6. The modified Figure 6 title is “Three dimensional structure decomposition of Pinus massoniana forest. (a): the RGB point cloud of Pinus massoniana forest samples;(b):the point cloud of Pinus massoniana forest 3D structure separation.”

We are very sorry for this low-level error. Similar issues have been checked and revised carefully.

(5)  For figure 7, please explain (a) to (d) in the figure caption, and move the explanation of pink area in the note or using legend.

Dear Reviewers: The relevant information is indicated in the title of Figure 7. The modified Figure 7 title is “Calculation of understory vegetation cover. (a): the result of the decomposition of the three-dimensional structure of the sample site; (b): the result of the reverse projection of the understory point cloud collection back to the orthophoto; (c): the two-dimensional image obtained after voxelization in Fig. 7b. In Figure 7c, the green part is the vegetation part, the black points are the understory vegetation points, and the pink area is the ground point .(d) :the result of the mask after binarization of the image. The black area represents the plant area, while the white area represents the bare land area”

(6)  Please reorganize the figure 8 and 9 as the comment above and explain (a) and (b) in the figure caption.

Dear Reviewer:The relevant information is indicated in the title of Figure 8 and figure 9. The modified Figure 8 title is “Linear regression results under different canopy closure.(a) the result of quantification of understory vegetation cover under low forest densities; (b):the result of quantification of understory vegetation cover under high forest densities.”

The modified Figure 9 title is “Linear regression results under different slope.(a): the result of quantification of understory vegetation cover under gentle slope conditions; (b): the result of quantification of understory vegetation cover under under inclined slope conditions; (c): the result of quantification of understory vegetation cover under steep slope conditions”

Point 13.   The authors mentioned “soil erosion” several times in this study. Is the prediction of understory vegetation cover only meaningful to soil erosion? Please clarify the meaning of this methodology or this study to remote sensing field. Could we use this method for other areas?

Dear Reviewer: The author mentions "soil erosion" several times in this paper because the National Foundation project that funded this paper hopes to obtain a series of stand factors such as understory vegetation cover to explore the forest quality and analyze the erosion control and restoration in the area. This paper proposes a remote sensing method for quantifying the extent of understory vegetation cover to provide a methodological and data base for subsequent studies, which will continue to fund research on the relationship between forest quality and "soil erosion". If the reviewer is interested in the research, you can check the results on the official website of the Ministry of Science and Technology of China after the end of this fund. This project will be closed around December 2022, and the follow-up work is continuing.

As a matter of fact, understory vegetation cover estimation is not only relevant for soil erosion. The Chinese government has decided to reach carbon peaking by 2030 and carbon neutrality by 2060. The forest carbon pool will play a large role in this strategy. In the forest ecosystem, not only the trees have the ability to store carbon elements, but also the understory vegetation can play its carbon storage capacity. Furthermore, information on the cover of understory vegetation is of great value in forest management, carbon cycling, and ecological research. As an example, in forest fire prediction, fire behavior models require height and cover of understory vegetation as inputs to detect fire-prone and fire-spreading areas; in studies of carbon and water cycling in large-scale boreal forest ecosystems, information on cover and species composition of mossy vegetation is essential for accurate estimation of net primary productivity of the entire ecosystem. We can use this method in other forestry fields. In the author's opinion, the innovation of this paper is to improve the technical process of near-ground active-passive remote sensing and to propose a high-precision decomposition method for the three-dimensional structure of forest stands. Colleagues can use the technical process of near-ground active-passive remote sensing to map the spectral information from passive remote sensing back to active remote sensing information, and can also use high-precision decomposition of the three-dimensional structure of the forest stand method to judge the health of the forest area (judging how many vegetation species and so on).

Point 14.   Some expression in this paper is confused or redundant. Please revise them. Here are a few examples:

(1)  Line 28 on page 1, it should be “southern hilly area of China”. You didn’t test it in other places.

Dear Reviewer: Very sorry for this error. The relevant content has been modified in the original article.

(2)  Line 35 on page 1, “hilly soil erosion area”? How to define it? Please revise it.

Dear Reviewer: The original text is“hilly soil erosion areas of southern China”. Since the topography of the study area is obviously undulating and surrounded by mountains, and the soil erosion in this area is very serious, the original content of this paper is “hilly soil erosion areas of southern China”.

(3)  Line 111 on page 3, do you mean “in hilly areas of southern China”?

Dear Reviewer: Yes, it does. The relevant content has been modified in the original article.

(4)  Line 200 on page 6, do you mean “two-dimensional visualization”?

Dear Reviewer: Yes. In the technical process of this study, the process of projecting the understory vegetation information enhanced by semantic segmentation to the UAV orthophoto using point cloud back projection can be considered as the understory vegetation point cloud visualized in two dimensions.

(5)  Please revise the title of section 4.2.

Dear Reviewer: The relevant title of 4.2 has been changed to “Results of three-dimensional structural decomposition of the stand in a sample plot of Pinus massoniana forest”.

(6)  Line 312 and 314 on page 10, why use “cloth simulation filter algorithm” on line 312, but “fabric filtering algorithm” on line 314? Are they two algorithms?

Dear Reviewer: Very sorry for this error. It is actually an algorithm, however, due to the English translation problem of this article, it is the case of a written error. The relevant content has been modified in the original article.

(7)  Line 315 on page 10, it should be a note in table 3.

Dear Reviewer: The relevant content has been transferred to the Table 4 table name as a comment for that table. The modified Table 4 table name is as follows.

Tab.4 Segmentation accuracy evaluation results.In the table, "❄/❄/❄" means "precision/recall/F1 index".

(8)  Line 365-366 on page 12, it should be moved to discussion secction.

Dear Reviewer: This content has been transferred in the first paragraph of section 5.2. The text Line 365-366 on page 12 in section 4.3 has been removed.

There are other mistakes and confusions. I cannot list them all. The manuscript should be polished by a native speaker.

In accordance with your comments, the main body of the article has been sought out for overall touch-ups and optimisation by a native English-speaking expert. The author has also focused on the advice given by professional retouching companies in the retouching process. So the revision lines may be slightly different.

Reviewer 3 Report

The work addresses an important theme and presents a new method to evaluate understory vegetation by combining active and passive RS data. However, the methodology is incomplete, lacking important information for understanding the processes carried out. The results are incomplete (missing Tables), described in a confusing way, and must be redone. As the results are vague, it was impossible to read the discussion.

The manuscript must be reconsidered.

Abstract

Line 12 - “southern hilly areas”. southern of china

Introduction

Line 49 - LiDAR? Change, Light Detection and Ranging (LiDAR).

Line 59 - “southern hilly region”. China?

Study area and sample site overview

Line 118 - What is the average altitude of the place?

Line 120 - “undulating terrain. Slope?

Line 126 - What is the RMSE of the coordinates collected by RTK? Inform.

Fig. 1 - Increase the font size of the map grid. Enlarge the part of the Figure with the vegetation detail (Tree).

Data and methodology

Line 146 - “processed and modeled”. With Ground Points Control (GCPs)? Inform.

UAV LiDAR data acquisition

Does the LiDAR UAV system have IMU, RTK? Inform.

What is the final precision (RMSE of X, Y and Z) of the generated point cloud?

Fig. 2 - Improve the quality.

Line 161 - “same coordinate system ”. Explain how this was accomplished.

Pre-processing date

Every topic must be redone. The algorithms used in LiDAR360 and ArcGIS Pro2.9 must be explained. The parameters used must also be informed.

“pre-processed by cropping and de-noising”. Explain how this is accomplished in LiDAR360.

“machine learning algorithm was used to pre-classify the point cloud into three categories: forest canopy, understory vegetation and ground points”. Explain how the algorithm works.

Results

Table 1 and 2. Where is it in the text?

Author Response

Response to Reviewer 3 Comments

Point 1.   Abstract:Line 12 - “southern hilly areas”. southern of china

Dear Reviewer: Very sorry for this error. The relevant content has been modified in the original article:changed line 12 from " southern hilly areas " to " southern hilly areas of China".

Point 2.   Introduction:

Line 49 - LiDAR? Change, Light Detection and Ranging (LiDAR).

Line 59 - “southern hilly region”. China?

Dear Reviewer: The relevant content has been modified in the original article: changed line 49 from " LiDAR " to " Light Detection and Ranging (LiDAR)". Meanwhile,we changed line 59 from " southern hilly region " to " southern hilly region of China". I am very sorry that there are so many low-level errors in this article.

Point 3. Study area and sample site overview

Line 118 - What is the average altitude of the place?

Line 120 - “undulating terrain. Slope?

Line 126 - What is the RMSE of the coordinates collected by RTK? Inform.

Fig. 1 - Increase the font size of the map grid. Enlarge the part of the Figure with the vegetation detail (Tree).

Dear Reviewer: According to the local government's statistical yearbook, the average altitude of the study area is 390 metres; the average slope of the hills in the study area is calculated to be around 20° by geostatistical analysis, and the average slope of the whole study area is around 7° without taking into account the influence of land use. The relevant content has been added in the original article.

According to the relevant parameters on DJI's official website, the RTK parameter used by the DJI Phantom 4UAS is D-RTK 2 (high precision GNSS mobile station). Comparing its technical manual, we can see that the RTK has an error of around 1cm in the X,Y plane and 3cm in the Z plane. The error was small and the results obtained allow for the next operation. The relevant content has been modified in the original article:” The four corner points of the sample plots were located by the DJI Phantom 4 RTK system with a positioning accuracy of centimetres.

For Fig. 1, we have increased the font size of the grid latitude and longitude in the updated Figure 1 and have included local forest photographs as part of the enlarged figure.

Point 4. Data and methodology

Line 146 - “processed and modeled”. With Ground Points Control (GCPs)? Inform.

Dear Reviewer: As the flight altitude was not high and the flight range was not large, only a 15m extension of the sample site was used as the flight range, so ground control points were not used in this study. Previous laboratory experiments have shown that at this level of accuracy, the use of ground control points to correct for ground height was limited.

Point 5.UAV LiDAR data acquisition

Does the LiDAR UAV system have IMU, RTK? Inform.

What is the final precision (RMSE of X, Y and Z) of the generated point cloud?

Fig. 2 - Improve the quality.

Line 161 - “same coordinate system ”. Explain how this was accomplished.

Dear Reviewer: According to the information provided by DJI's official website, DJI M300 integrated highly integrated, LiDAR, mapping camera and high precision inertial guidance, while M300 RTK can provide up to 1cm+1ppm positioning accuracy in plane and 1.5cm+1ppm in elevation, which is composed of GNSS and INS together to form a high precision positioning and attitude measurement system, which can provide high precision process data files for post-processing results. Therefore, the relevant content has been modified in the original article:” The M300's built-in RTK provides a maximum positioning accuracy of 1cm+1ppm on plane and 1.5cm+1ppm on elevation, high-precision data files were obtained with the help of GNSS, INS, IMU and inertial guidance system.

The part of actual ground measurements and the part of joint active-passive remote sensing to quantify the understory vegetation and analyze it are added to the original diagram.Relevant figures have been modified in the paper. The modified results are as follows.

The image stitching software developed by DJI, DJI Terra, has the function of projecting remote sensing images with point clouds. This study uses this function to assign the same projection information to both the stitched orthophoto and the processed point cloud information, so that both are in the same coordinate system. To avoid creating misunderstandings, the relevant content has been modified in the original article:” In advance, the UAV visible photos and airborne LiDAR data of the Pinus mas-soniana forest sample sites were fetched, and the remote sensing data were ensured to be within the WGS_1984_UTM_Zone_50N geographic coordinate frame.”

Point 6.Pre-processing date

Every topic must be redone. The algorithms used in LiDAR360 and ArcGIS Pro2.9 must be explained. The parameters used must also be informed.

Dear Reviewer: When training a PointCNN model using arcgis.learn, the raw point cloud dataset in LAS files was first converted into blocks of points, containing a specific number of points along with their class codes.For this step of exporting the data into an intermediate format, use Prepare Point Cloud Training Data tool. The training related codes are as follows.

import arcpy

arcpy.env.workspace = 'D:/data'

arcpy.ddd.PreparePointCloudTrainingData('training_forest.lasd', '800 Meters',

'vegetation_training.pctd', 'validation_point_cloud='validation_source.lasd',

                   class_codes_of_interest=[0, 2,3,5], block_point_limit=8096,

)

After training the PointCNN model, each type of metric (precision, recall and f1-score) associated with the validation data was calculated and the model was saved.

For inferencing, use Classify Points Using Trained Model tool, in the 3D Analyst extension, available from ArcGIS Pro 2.8 onwards. The training related codes are as follows.

import arcpy

arcpy.env.workspace = 'D:/data/'

arcpy.ddd.ClassifyPointCloudUsingTrainedModel('202206_forest.lasd', 'forest_infrastructure_classification.emd',

[0, 2, 3, 5], 'EDIT_SELECTED')

To ensure that information was not missed, add the following to the original text:

“The LiDAR data were processed first using LiDAR360 (https://www.lidar360.com/) software. The acquired LiDAR point cloud raw data were pre-processed by cropping and de-noising, while the software's built-in machine learning algorithm was used to pre-classify the point cloud into three categories: forest canopy, understory vegetation and ground points. The software's machine learning classification tool uses random forests to classify the point cloud data. By manually editing the categories of a typical of data in the same batch, the model was trained and then batched to process a large amount of data. The classification results will be further processed to construct a semantic segmentation dataset.

Hence, the preliminary classified point cloud data were loaded into the ArcGIS Pro2.9 deep learning environment and divided into 800 points×800 points tiles. Due to graphics card memory limitations, the batch size was set to 4. The training categories re-main forest canopy, understory vegetation and ground points. The point cloud was then finely tagged and classified using the point cloud tagging function, and the semantic segmentation dataset was built based on the fine classification results. Finally, the tag classification results were corrected by a visual interpretation check.”

“pre-processed by cropping and de-noising”. Explain how this is accomplished in LiDAR360.

Dear Reviewer: The LiDAR360 related toolbar is shown below.

Common noises include high level slop and low level slop. High level slop is usually caused by the airborne LiDAR system being affected by low flying objects (such as birds or aircraft) during data acquisition and mistakenly recording the reflected signal from these objects as the reflected signal from the target being measured. Low-level coarse deviations, on the other hand, are extreme lows caused by multi-path errors in the measurement process or errors in the laser rangefinder. By choosing the right parameters, the noise can be removed and the quality of the data improved.

Assuming a standard deviation multiplier of meanK, the algorithm will search for each point for a specified number of neighbouring points, calculate the average of the distances D from the point to the neighbouring points, calculate the median of these average distances meanD and the standard deviation S. If D is greater than the maximum distance MaxD (MaxD = meanD + meanK * S), it is considered a noise point and will be removed. The relevant drawings are as follows. Default parameters are selected for the parameters.

The software also provides a crop-by-rectangle tool that allows you to draw rectangular ranges based on multiple rectangular ranges entered by the user or interactively with the view, extracting all the point cloud data within each rectangular range and saving it in one file or multiple files.

“machine learning algorithm was used to pre-classify the point cloud into three categories: forest canopy, understory vegetation and ground points”. Explain how the algorithm works.

Dear Reviewer: The original forest LiDAR point cloud data contains the spatial position of each point and sometimes also information on attributes such as reflection intensity, echo counts, RGB etc. However, in order to perform more in-depth analysis and applications of forest point cloud data, it was first necessary to classify it. LiDAR360 provides a very comprehensive set of classification tools, including automatic and interactive classification. This interactive classification method manually classifies a small number of ground and vegetation points in the data, and then uses this sample data to classify all sample data by machine learning classification.

Click to change the point cloud display mode to RGB for better feature identification. In this tutorial, the target categories are: ground points, high vegetation and low vegetation. Other points will be classified as unclassified. The interactive classification tool was in the profile editing tool of the LiDAR360 software.

1) In the layer management window, only the sample data was displayed. Open the profile editing toolbar.

2) High vegetation is relatively easy to distinguished, so divided the high vegetation first. Click on , in the profile editing window, to draw the extent of the profile area along the ground. You can click on the drop-down menu in the menu bar to switch between different views of the point cloud in the profile window, or click on the rotate button to rotate the point cloud in the profile window. In the Method panel of the profile editing window, clicked on Category Settings and set the target category to High Vegetation Points.

3) Use the polygon selection tool to select the highly vegetated points and double click the left mouse button to complete the selection.

See the picture above

4) Use the same method to split the ground points and low vegetation points and determine the sample.

The LiDAR360 software's machine learning classification tool uses random forests to classify the point cloud data. By manually editing the categories of a small amount of data in the same batch, the model was trained and then batched to process a large amount of data.

Point 7.Results

Table 1 and 2. Where is it in the text?

Dear Reviewer: Table 1 in the original paper represents typical conditions for each canopy closure condition sample plot, and Table 2 represents typical conditions for each slope condition sample plot. Since the information represented in Figure 4 was transferred to Table 1, it is now Tables 2 and 3. After modification, Table 2 and Table 3 are shown below.

  Tab.2 Classification Canopy closure results of typical sample plots

Canopy closure case

(Digital Surface Model)

Legend

Low canopy closure

(0.1-0.4)

High canopy closure

(0.4-0.7)

Tab.3 Classification Slope results of typical sample plots

Topographic profile (Point cloud)

Legend

Gentle Slopes

(6°-12°)

Inclined Slopes

(13°-22°)

Steep slopes

(>22°)

In accordance with your comments, the main body of the article has been sought out for overall touch-ups and optimisation by a native English-speaking expert. The author has also focused on the advice given by professional retouching companies in the retouching process. So the revision lines may be slightly different.

Round 2

Reviewer 2 Report

Comments to authors:

The manuscript has been greatly improved. It could be accepted after the following comments are considered.

Minor revisions:

1.     I don’t recommend to show table 1 like this. It is too long. Additionally, it is not “statistics”, just information for each plot. Statistics should be minimum, maximum, mean, std, … of the three variables of all plots.

2.     The title of section 4.2 should be shortened. Please consider “Three-dimensional structural decomposition” or “Three-dimensional structural decomposition of Pinus massoniana forest”.

3.     Some words in figure titles are redundant. Please recheck and revise them. For example, for figure 7 title, it is better to use:

“Linear regression of ground and estimated UVC under different canopy closure:(a) low forest densities; (b) high forest densities”

Please check all the figure titles in the manuscript.

4.     Line 495-496 on page 16, please revise the sentence. It is not clear enough. `The cost was greater than what?

Author Response

Response to Reviewer 2 Comments

Point 1. I don’t recommend to show table 1 like this. It is too long. Additionally, it is not “statistics”, just information for each plot. Statistics should be minimum, maximum, mean, std, … of the three variables of all plots.

Dear Reviewer: Based on your comments, we have made the following changes:Restore the content of the original Figure 4 and add the sample plot data statistics table to it.

                       Fig.4 Canopy closure and topographic distribution of the sample plot

Tab.1 Statistical results of sample site factor information

UVCGround

Average slope(°)

Canopy closure

mean

0.539

16.466

0.36

min

0.301

5.341

0.1

max

0.755

27.596

0.6

std

0.124

4.791

0.119

Point 2. The title of section 4.2 should be shortened. Please consider “Three-dimensional structural decomposition” or “Three-dimensional structural decomposition of Pinus massoniana forest”.

Dear Reviewer: After summarizing the content of this subsection, the title of 4.2 has been amended to read“Three-dimensional structural decomposition of Pinus massoniana forest”, in order to ensure that the title is concise but not inaccurate.

Point 3. Some words in figure titles are redundant. Please recheck and revise them. For example, for figure 7 title, it is better to use:

“Linear regression of ground and estimated UVC under different canopy closure:(a) low forest densities; (b) high forest densities”

Please check all the figure titles in the manuscript.

Dear Reviewer: Thank you for your advice! The name of Figure 7 has been amended to Linear regression of ground and estimated UVC under different canopy closure:(a) low forest densities; (b) high forest densities”. Also, based on your suggestion, the name of Figure 8 has been amended to Linear regression results under different slope.(a): gentle slope conditions; (b): inclined slope conditions; (c): steep slope conditions”.

Point 4.Line 495-496 on page 16, please revise the sentence. It is not clear enough. `The cost was greater than what?

Dear Reviewer: Thank you for your advice!The second paragraph of 5.4 as a whole has been merged with the second paragraph of 5.3 in accordance with the logic of the article. The main point of the paragraph was that the deep learning algorithm proposed in this paper is more expensive to use than traditional machine learning algorithms such as random forests. Meanwhile, the cost of acquiring LiDAR point clouds is also greater than the cost of acquiring UAV multispectral data.

Based on this, the following changes have been made to the paper:

Add in Line 488-495

“However, the cost of using the deep learning algorithms used in this study was significant relative to traditional machine learning methods that can be used in forested situations. In order to obtain proper results, a highly configured graphics card with high graphics memory and almost two days of uninterrupted training were re-quired to obtain a constructed analysis result. Furthermore, the cost of acquiring point clouds using airborne LiDAR was also greater than that of acquiring point clouds using multispectral drones.And it is currently not possible to use relatively simple UAV point clouds for training and decomposition of forest 3D structures.”

Reviewer 3 Report

Congratulations to the authors.

Author Response

Point 1. Congratulations to the authors.

Dear Reviewer: Thank you for your sincere advice! Your pointers are a way forward for the writer and an important factor in improving the quality of the paper. We notice, however, that there is room for negotiation in your citation of references, so we cite the following[1-3], based on a thorough reading of the article and an understanding of the a priori research.

  1. Zhang, W.; Gao, F.; Jiang, N.; Zhang, C.; Zhang, Y. High-Temporal-Resolution Forest Growth Monitoring Based on Segmented 3D Canopy Surface from UAV Aerial Photogrammetry. Drones. 2022, 6, 158.
  2. Zhang, Y.; Onda, Y.; Kato, H.; Feng, B.; Gomi, T. Understory biomass measurement in a dense plantation forest based on drone-SfM data by a manual low-flying drone under the canopy. J. Environ. Manage. 2022, 312, 114862.
  3. Liu, X.; Su, Y.; Hu, T.; Yang, Q.; Liu, B.; Deng, Y.; Tang, H.; Tang, Z.; Fang, J.; Guo, Q. Neural network guided interpolation for mapping canopy height of China's forests by integrating GEDI and ICESat-2 data. Remote Sens. Environ. 2022, 269, 112844.
